# MiR 208a Regulates Mitochondrial Biogenesis in Metabolically Challenged Cardiomyocytes

**DOI:** 10.3390/cells10113152

**Published:** 2021-11-13

**Authors:** Naveen Mekala, Jacob Kurdys, Alexis Paige Vicenzi, Leana Rose Weiler, Carmen Avramut, Edwin J. Vazquez, Neli Ragina, Mariana G. Rosca

**Affiliations:** Department of Foundational Sciences, Central Michigan University College of Medicine, Mount Pleasant, MI 48858, USA; naveen.mekala@temple.edu (N.M.); kurdysj@gmail.com (J.K.); vicen1ap@cmich.edu (A.P.V.); weile1lr@cmich.edu (L.R.W.); avram1c@cmich.edu (C.A.); edwinjvazquez@gmail.com (E.J.V.); ragin1n@cmich.edu (N.R.)

**Keywords:** cardiomyocytes, metabolic syndrome, miR 208a, mitochondrial biogenesis, bioenergetics

## Abstract

Metabolic syndrome increases the risk for cardiovascular disease including metabolic cardiomyopathy that may progress to heart failure. The decline in mitochondrial metabolism is considered a critical pathogenic mechanism that drives this progression. Considering its cardiac specificity, we hypothesized that miR 208a regulates the bioenergetic metabolism in human cardiomyocytes exposed to metabolic challenges. We screened in silico for potential miR 208a targets focusing on mitochondrial outcomes, and we found that mRNA species for mediator complex subunit 7, mitochondrial ribosomal protein 28, stanniocalcin 1, and Sortin nexin 10 are rescued by the CRISPR deletion of miR 208a in human SV40 cardiomyocytes exposed to metabolic challenges (high glucose and high albumin-bound palmitate). These mRNAs translate into proteins that are involved in nuclear transcription, mitochondrial translation, mitochondrial integrity, and protein trafficking. MiR 208a suppression prevented the decrease in myosin heavy chain α isoform induced by the metabolic stress suggesting protection against a decrease in cardiac contractility. MiR 208a deficiency opposed the decrease in the mitochondrial biogenesis signaling pathway, mtDNA, mitochondrial markers, and respiratory properties induced by metabolic challenges. The benefit of miR 208a suppression on mitochondrial function was canceled by the reinsertion of miR 208a. In summary, miR 208a regulates mitochondrial biogenesis and function in cardiomyocytes exposed to diabetic conditions. MiR 208a may be a therapeutic target to promote mitochondrial biogenesis in chronic diseases associated with mitochondrial defects.

## 1. Introduction

Metabolic syndrome, frequently induced by a high caloric intake, is the prerequisite for type 2 diabetes (T2D) and increases the risk for cardiovascular diseases including metabolic cardiomyopathy [1,2] in both humans [3,4] and animal models [5,6]. Cardiac abnormalities start early in obese subjects and progress to increased left-ventricular (LV) mass [7,8] causing reduced diastolic compliance and filling [9] (heart failure with preserved ejection fraction), followed by systolic dysfunction and congestive heart failure. The prevalence of cardiomyopathy is approximated to be 50% among diabetic human subjects [10,11], most of them presenting with diastolic dysfunction [3,4,12,13].

Pathogenic mechanisms leading to metabolic cardiomyopathy remain controversial and include metabolic challenges (hyperglycemia and hyperlipidemia), insulin resistance and hyperinsulinemia, activation of the renin–angiotensin system, inflammation, epigenetic changes, oxidative stress, and abnormalities in cardiac bioenergetics. All these mechanisms lead to cardiac remodeling with hypertrophy, apoptosis, and fibrosis [14]. Alterations in cardiac oxidative metabolism are considered critical pathogenic mechanisms for metabolic cardiomyopathy.

Fatty acids are the major carbon fuels in normal heart. Normal cardiac metabolism exhibits physiologic shifts between glucose and fatty acids (FA) as energetic substrates. In high-fat diet (HFD)-induced metabolic syndrome with glucose intolerance, although exposed to an excess of energetic substrates, the heart becomes almost completely reliant on FA as bioenergetic fuels for ATP generation. To be efficient, this metabolic change must be supported by coupling the increased FA oxidation with ATP generation. Fatty acids are catabolized to acetyl-CoA feeding the tricarboxylic acid cycle (TCA) and the reducing equivalents, NADH and FADH2, which are further oxidized by the electron transport system (ETS) to form ATP. It is reported that, during lipid excess, the heart, in comparison with skeletal muscle, does not accumulate intermediate FA oxidation compounds (i.e., FA-derived acylcarnitine species) [15], indicating that mitochondrial FA oxidation is complete. Therefore, an enhanced FA oxidation resulting in ATP generation must be supported by a coordinated activation of FA β-oxidation pathway, TCA cycle, and ETS. Metabolic remodeling toward increased mitochondrial FA β-oxidation is controlled by mitochondrial and nuclear mechanisms that govern the biogenesis of mitochondrial components. Despite this early activation, mitochondrial dysfunction is considered a major pathogenic mechanism driving the progress of metabolic cardiac disease to heart failure [16].

MicroRNAs (miRs) belong to a family of single-stranded small RNAs that regulate cardiac function in both health and disease [17] by suppressing protein synthesis via either inhibiting mRNA expression or initiating mRNA degradation [18]. MiRs target a broad spectrum of signaling pathways including bioenergetics, oxidative stress, calcium handling, and apoptosis [18]. The diabetic heart experiences either upregulation or downregulation of various miRs [19], including the muscle-specific miR-1 [20] and cardiac-specific miR 208a [20].

MiR 208a is produced by intron 27 of the human and mouse *myh6* gene that encodes the α isoform of the myosin heavy chain (α-MHC) and is co-transcribed in parallel with its host gene during normal cardiac development [21]. MiR 208a expression does not correlate with the expression of its host gene, *myh6,* in pathological conditions. For example, in a model of transaortic constriction, *myh6* gene expression is decreased while miR 208a is unchanged. Genetic deletion of the miR 208a in mice does not cause an overt phenotype in basal conditions while impeding the cardiac response to different types of stress, including pressure overload and hypothyroidism. Cardiomyocytes lacking miR 208a fail to upregulate the *myh7* gene encoding the β-MHC contractile protein [21], indicating that miR 208a is required for cardiac remodeling and contractile function. 

MiR 208a was reported increased in diabetic human hearts [22] and presented an oscillated pattern in diabetic mouse hearts with an early increase followed by a drop in advanced diabetes [22]. Pharmacologic inhibition of miR 208a improved cardiac and systemic insulin sensitivity and glucose metabolism in obese and diabetic mouse models [23]. However, the role of miR 208a in cardiac mitochondrial function during metabolic challenges has not been investigated. The objective of our study was to determine the role of the cardiac specific miR 208a in regulating mitochondrial bioenergetics in metabolically challenged human cardiomyocytes. Our major findings were that miR 208a is decreased after short-term exposure to diabetic conditions (high glucose, high albumin-bound palmitate). MiR 208a downregulation prevented changes in cardiac contractile proteins and preserved mitochondrial biogenesis and function in cardiomyocytes incubated in diabetogenic conditions. We conclude that miR 208a is a negative regulator of mitochondrial oxidative metabolism.

## 2. Materials and Methods

### 2.1. Reagents

Unless otherwise mentioned when detailing specific methods, all reagents were purchased from Sigma-Aldrich and were of the highest purity grade.

### 2.2. Cells

Immortalized human adult LV cardiomyocytes, SV40 (Applied Biological Materials, Inc, accessed on 20 January 2019) were cultured in Prigrow medium (TM001, provided by the company) supplemented with 10% fetal bovine serum, 100 U/mL penicillin, and 100 µg/mL streptomycin (Thermo Fisher, Waltham, MA, USA).

To suppress miR 208a, we used a clustered regularly interspaced short palindromic repeat/Cas9 (CRISPR) editing approach in collaboration with Synthego (synthego.com, accessed on 26 March 2019). The editing efficiency was approximately 90%. All experiments were performed with cells at 60–70% confluence. For transfection, 10,000 cells/well were seeded on the eight-well plate of the Agilent Seahorse XFp instrument, incubated overnight, and subsequently transfected with 3 pmol of miR 208a mimic (mirVana, hsa-miR-208a-3p, MC10677, sequence AUAAGACGAGCAAAAAGCUUGU)/100 µL using Lipofectamine (0.5 µL/100 uL) for 24 h according to the manufacturer’s protocol. Scrambled sequence was used as miR mimic negative control. After 24 h, transfection medium was replaced with incubating media. To mimic diabetogenic conditions, incubating media contained 25 mM glucose and 20 µM bovine serum albumin-bound palmitate (BSA-bound palmitate) prepared as described [24] in Dulbecco’s modified Eagle’s medium (DMEM) supplemented with 10% fetal bovine serum, 100 U/mL penicillin, and 100 µg/mL streptomycin, under humidified air with 5% CO_2_ at 37 °C.

### 2.3. Respiratory Studies in Cultured Cardiomyocytes

SV40 cardiomyocytes were plated on the eight-well plate of the XFp analyzer (Agilent Technologies), grown to 60–70% confluence, and incubated for 24 h with either DMEM (normal conditions, N), high glucose (HG, 25 mM), or high palmitate (HP, 20 µM, 5:1 BSA: palmitate ratio). A pilot experiment was performed to determine the optimal number of cardiomyocytes that linearly correlates with basal oxygen consumption rates (OCR). Therefore, respiratory experiments were performed with a seeding density of 20,000 cells/well for wildtype and 15,000 cells/well for the miR 208a-deficient cardiomyocytes. Respiratory properties were assessed in bicarbonate- and phenol red-free DMEM supplemented with energetic substrates (10 mM glucose, 1 mM pyruvate, 2 mM glutamine). After basal oxygen consumption and extracellular acidification rates (OCR and ECAR, respectively) were measured, cells were sequentially challenged with oligomycin (ATP synthase inhibitor, 25 µM), FCCP (carbonyl cyanide-*p*-trifluoromethoxyphenylhydrazone, mitochondrial uncoupler, 5 µM), rotenone (complex I inhibitor, 5 µM), and antimycin A (complex III inhibitor, 5 µM). Different concentrations of oligomycin (2.5–25 µM) and FCCP (2.5–10 µM) were used in preliminary experiments to obtain the maximal effects on OCRs.

Basal OCR was calculated by subtracting the non-mitochondrial OCR (minimal OCR after complete ETC inhibition with rotenone and antimycin A) from the OCR value prior to FCCP injection. Oligomycin-sensitive ATP-coupled OCR was calculated by subtracting the oligomycin-induced OCR from the basal OCR. The OCR needed to overcome the proton leak across the mitochondrial inner membrane was calculated by subtracting the non-mitochondrial OCR from the oligomycin-induced OCR. Mitochondrial maximal OCR was calculated by subtracting the non-mitochondrial OCR from the OCR induced by FCCP injection. Reserve (spare) respiratory capacity was calculated by subtracting basal OCR from maximal OCR.

Extramitochondrial glycolysis was assessed as extracellular acidification rate (ECAR). The last ECAR value before oligomycin addition was defined as basal ECAR. Reserve ECAR was calculated by subtracting basal ECAR from the oligomycin-induced ECAR. All OCR and ECAR values were expressed per mg protein measured by the Lawry assay.

### 2.4. Gene Expression Analysis

Predicted miR 208a target genes were determined using v7.0 of the TargetScan database (http://www.targetscan.org/cgi-bin/targetscan/vert_71/targetscan.cgi accessed on 20 August 2019). From the predicted target genes, only those with mirsvr score <−0.1 and potential to regulate mitochondrial function were included for further analyses [25].

### 2.5. Real-Time Reverse Transcription PCR (RT-PCR), and Quantitative PCR

Total RNA and microRNA-enriched RNA were isolated from cultured cardiomyocytes with mirVana RNA isolation kit (Thermo Fisher, AM1561) using the manufacturer’s instructions and a described protocol [26]. 

Total RNA (2 µg/sample) was used to generate cDNA using Super Script II reverse transcriptase kit (Invitrogen, Carlsbad, CA, USA). RT-PCR was performed using primers presented in Appendix A. RT-PCR cDNA products were used as templates for quantitative PCR (qPCR) assays, which were run in duplicates on StepOnePlus Real-Time PCR System using SYBR green PCR master mix (Invitrogen). Real-time PCR cycling conditions consisted of 95 °C for 10 min, followed by 40 cycles of 95 °C for 15 s and 60 °C for 1 min. RT-PCR products were separated on 1.5% agarose gels, while the bands were quantified by densitometry using the LI-COR Odyssey imaging system and expressed relatively to the housekeeping genes *U6* and *β-actin.*

### 2.6. Western Blot Experiments

Cells (3 × 10^5^/well) were seeded on six-well plates, incubated for 24 h, and lysed. Denatured proteins (20 µg) were separated on 4–12% Tricine gels (Invitrogen), electroblotted onto PVDF membranes, and probed with primary (1:1000 dilution) and LI-COR infrared-based secondary antibodies (1:5000 dilution). Protein bands were quantified by densitometry using an LI-COR image scanner. A complete list of antibodies is provided in Appendix A.

### 2.7. Mitochondrial DNA

Total DNA was isolated from cultured cardiomyocytes (DNA Purification System, Promega, Madison, WI, USA), quantified by a spectrophotometer (NanoDrop, Thermo Scientific, Waltham, MA, USA), and subjected to quantitative real-time PCR using designed primers for mitochondrial NADH dehydrogenase 4 (forward: 5′–CAGCCACATAGCCCTCGTAG–3′; reverse: 5′–GCGAGGCTTGCTAGAAGTCA–3′). To increase the rigor of our research, we also used an alternative primer for mitochondrial NADH dehydrogenase subunit 4 (forward: 5′–TCCTCCCTACTATGCCTAG–3′; reverse: 5′–AGCATTCGGAGACAACAG–3′).

### 2.8. Statistics

Statistical analysis was performed using GraphPad Prism 7. For multiple comparisons, we used one- or two-way ANOVA. A paired two-tailed *t*-test was performed for pairwise comparisons. A *p*-value < 0.05 was deemed significant. The graphs show individual data, with mean ± SEM.

## 3. Results

### 3.1. MicroRNA 208a in Human Cardiomyocytes under Metabolic Stress

Using RT-PCR, we found that incubation of the human cardiomyocytes SV40 in diabetogenic conditions (25 mM glucose, 20 mM BSA bound palmitate) caused a time-dependent decrease, up to 30%, in miR 208a expression (Figure 1).

### 3.2. Predicted Targets of MiR 208a

We performed a manual search to predict the mRNA species targeted by miR 208a, followed by confirmatory RT-PCR. We found that 11 genes scored as the strongest predicted targets with the TargetScan program: *Arl16* (ADP ribosylation factor-like GTPase 16), *Stc1* (Stanniocalcin 1, *Ncoa7* (nuclear receptor coactivator 7), *ATP5* (mitochondrial ATP synthase subunit 5), *Tomm6* (translocase of outer membrane 6), *MED7* (mediator complex subunit 7), *Degs1* (sphingolipid delta(4)-desaturase 1), *Snx10* (Sortin nexin 10), *Stard4* (StAR-related lipid transfer domain-containing 4), *MRPS28* (mitochondrial ribosomal protein S28), and *Hmgn3* (high-mobility group nucleosomal-binding domain 3).

To confirm if miR 208a controls these mRNAs in normal and diabetic conditions, we used CRISPR editing to eliminate approximately 90% of miR 208a in SV40 human cardiomyocytes (Figure 2A). The exposure of cardiomyocytes to HP and HG + HP downregulated *STC1* mRNA, which was reversed by the miR 208a deficiency (Figure 2B). *MED7* mRNA was also significantly downregulated by all diabetogenic conditions and rescued by the miR 208a deficiency (Figure 2C). The *SNX10* mRNA was severely downregulated by HG, upregulated by HP incubation, and upregulated by the miR 208a-deficient state in both normal and HG conditions (Figure 2D). The mRNA encoding for mitochondrial ribosomal protein S28 (MRSP28) was severely downregulated by the full metabolic milieu (HG + HP) in SV40 wildtype cardiomyocytes and completely recovered by the miR 208a deficiency, whereas it was upregulated by the miR 208a-deficient state in normal incubating conditions (Figure 2E). The mitochondrial contact site and cristae organizing system protein 1 (MICOS1, MINOS1) was decreased by all diabetogenic conditions and not affected by the miR 208a status (Figure 2F). In summary, *SNX10* and *MRSP28* mRNAs were increased in miR 208a-deficient cardiomyocytes cultured in normal conditions. STC1, MED7, SNX10, and MRSP28 mRNAs were depressed by diabetic conditions and reversed by miR 208a deficiency. MINOS1 was insensitive to the miR 208a status.

### 3.3. MiR 208a Suppression Protects against the Myosin Isoform Switch and Pathological Stress Markers Induced by Metabolic Challenges

MiR 208a suppression increased MHC-β and decreased ANP protein expression in normal (N) conditions (Figure 3). The decrease in MHC-α and increase in MHC-β induced by HG and HP were reversed by miR 208a downregulation. Double HG + HP exposure did not have a summative effect on either contractile protein. Both diabetogenic conditions increased the expression of ANP, which was reversed by the miR 208a downregulation.

### 3.4. MiR 208a, Mitochondrial Biogenesis, and Mitochondrial Markers

MiR 208a suppression did not alter the protein level of any mitochondrial biogenesis factors in normal incubating conditions (Figure 4A). TRA protein level was unchanged by either HG or HP, whereas it increased upon miR 208a suppression (Figure 4A). TFAM and NRF1 were decreased by diabetic conditions and reversed by miR 208a suppression. NRF1 and PGC1α proteins were decreased by HG and reversed by miR 208a suppression (Figure 4A,B). In agreement with TFAM, mtDNA was significantly decreased by diabetic conditions and normalized by miR 208a suppression (Figure 4C and Appendix A).

We verified whether changes in mtDNA are accompanied by similar alterations in mitochondrial markers (Figure 4D,E), and we found that ETS subunits are not affected by the miR 208a status in normal conditions. In contrast, diabetic conditions downregulated subunits of complexes I (NDUFB8, 20 kDa), III (core protein 2, 48 kDa), and V (α-subunit, 55 kDa), which was reversed by miR 208a suppression (Figure 4D). Mitochondrial FA oxidation enzymes, CPT1, and long-chain acylCoA dehydrogenase (LCAD) followed a similar pattern as the mitochondrial ETS markers. In contrast, cytochrome c amount was increased by the diabetogenic conditions and not affected by miR 208a status. In summary, while mitochondrial bioegenesis signal was inconsistently affected by diabetic conditions and miR 208a status, mtDNA and markers, except cytochrome c, were decreased by diabetic conditions and reversed by miR 208a deficiency.

### 3.5. MiR 208a Suppression Protects against Mitochondrial Dysfunction Induced by the Metabolic Stress

MiR 208a suppression caused a slight decrease in the basal OCR (Figure 5A) and an increase in the reserve OCR, while other respiratory parameters were unchanged. Changes in OCR were mirrored by an increase in the basal ECAR and a decrease in reserve ECAR (Figure 5A). While miR 208a overexpression slightly decreased the basal OCR in normal conditions, the other respiratory parameters were unchanged (Figure 5B).

The effect of miR 208a on cardiomyocytes cultured in high glucose (HG) is shown in Figure 5C. Both basal and proton leak OCR were decreased by miR 208a suppression upon HG incubation and not reversed by miR 208a re-expression (Figure 5C). ADP-dependent and non-mitochondrial OCRs were not affected by the miR 208a suppression. In contrast, maximal and reserve OCRs were decreased by HG, reversed by miR 208a suppression, and depressed by miR 208a re-expression (Figure 5C).

Basal and proton leak OCRs were decreased by miR 208a suppression during HP incubation and not reversed by miR 208a re-expression (Figure 5D). ADP-dependent OCR was not affected by the miR 208a status. In contrast, maximal and reserve OCRs were decreased by HP, reversed by miR 208a suppression, and depressed by miR 208a re-expression (Figure 5D). HP incubation decreased the non-mitochondrial oxygen consumption, which was reversed by the miR 208a suppression and negatively affected by miR 208a re-expression.

Appendix A shows the effect of the scrambled control miR sequence on SV40 cardiomyocytes respiratory properties. We confirmed that the basal OCR is decreased by the miR 208a deficiency, and that the transfection with scrambled miR has no effect on basal OCR. We also confirmed that miR 208a suppression prevented the HG- and HP-induced decrease in maximal and reserve OCRs, and that scrambled miR transfection did not affect these rates. A full description of the effect of miR 208 overexpression on mitochondrial respiratory properties is provided in Appendix A. We show that miR 208a overexpression decreased basal and proton leak OCR in all three experimental conditions. We confirmed that the exposure to HG and HP decreased the maximal and reserve OCR; however, the miR 208a overexpression did not have an additive effect on the collapse of the maximal and reserve OCR.

Basal OCR positively correlated with ADP-coupled OCR, and they both negatively correlated with the reserve OCR in all experimental conditions disregarding the miR 208a status (Appendix A). While basal OCR positively correlated with the maximal OCR, miR 208a suppression induced a weak negative correlation in normal conditions. Maximal OCR positively correlated with ADP-coupled OCR only in HP conditions with mir 208a suppression, strengthening this correlation, and it exhibited a strongly positive correlation with the reserve OCR in HG condition disregarding the 208a status.

In summary, the maximal and reserve OCRs were decreased by the diabetic conditions and reversed by the miR 208a deficiency in SV40 human cardiomyocytes.

## 4. Discussion

This report focuses on the effect of miR 208a on mitochondrial bioenergetics in metabolically challenged human cardiomyocytes. Using loss- and gain-of-function experiments, we show that miR 208a regulates the cardiac mitochondrial stress response during metabolic challenges while having minimal effects in homeostatic conditions. Our data indicate that miR 208a suppression protects mitochondrial integrity during metabolic heart disease and suggest that miR 208a may be a therapeutic target in hereditary and acquired mitochondrial diseases.

### 4.1. MiR 208a in Cardiomyocytes Exposed to Metabolic Stress

We observed that both metabolic challenges (HG and HP) equally downregulated miR 208a by 30% in a time-dependent manner in cultured cardiomyocytes. In contrast, in chronic diseases, cardiac miR 208a expression varies over the course of the disease. For example, in obese db/db mice, miR 208a exhibited an oscillatory pattern with an early increase followed by a later decline [22]. MiR 208a is a stable microRNA, with a half-life longer than 12 days [21]; therefore, our short (24 h) experimental conditions may not have depicted miR 208a changes larger than 30% and its potential oscillations during metabolic challenges.

### 4.2. Predicted MiR 208a Targets

MiR 208a may cause either degradation or translational repression of targeted mRNAs [27]. We used bioinformatic tools to predict the miR 208a mRNA targets by pairing the miR 208a seed region with complementary sites within mRNAs. Both mRNA canonical sites, containing the exact partner bases in the miR 208a seed region, and noncanonical sites, without a complete match with the miR seed, were examined. Because some canonical sites are more functionally efficient than others to cause protein repression, in order to determine the miR 208a targeting efficiency, we used a quantitative approach taking into account multiple features including the miR 208a target site type and mRNA 3′-UTR terminus characteristics. These features were combined to develop the total context score (TCS) [28]. We then confirmed the effect of miR 208a deficiency by assessing the expression of targeted mRNAs. We found that two mRNAs are increased by miR 208a suppression in the absence of the cardiomyocyte stress, Snx10 and MRPS28, with the most significant TCS of −1.01 and −0.70, respectively.

Our results indicate that the Snx protein family is affected by metabolic stress, and they suggest that miR 208a reverses the alterations of the metabolic pathways regulated by the Snx family. Snx10 belongs to the Sortin nexin family that regulates protein trafficking and recycling. Snx10 supports energetic metabolism via limiting chaperone-mediated lysosomal autophagy (CMA)-dependent degradation of glycolysis and tricarboxylic acid cycle enzymes [29]. The 3′-UTR terminus of the human Snx10 mRNA has three sites that can potentially bind a nucleotide sequence in the miR 208a seed. Our respiratory studies showing a depression by metabolic challenges and reversal by the miR 208a suppression are in line with the concept that an intact Snx10 limits CMA and may favor mitochondrial metabolism.

We show that MRPS28 mRNA is decreased by the complete metabolic milieu (HG + HP) and normalized by miR 208a suppression. MRPS28 mRNA generates the small protein subunit bS1m, an essential component of the mitochondrial ribosome [30], and it has one site in its 3′-UTR terminus that can potentially bind a nucleotide sequence in the miR 208a seed. MRPS28 deficiency inhibited mitochondrial translation, thus decreasing mitochondrial biogenesis and respiration [31], confirming that MRPS28 is critical for mitochondrial integrity. Our data indicate that miR 208a regulates the mitochondrial translation in cardiomyocytes during metabolic stress.

Stanniocalcin-1 (Stc1) mRNA has two sites in the 3′-UTR terminus, which can bind a nucleotide sequence within the miR 208a 3p terminus located outside of the seed sequence. Stc1 mRNA was not affected by the miR 208a status in the absence of metabolic stress, whereas it was decreased by HP and reversed by miR 208a deficiency. Stc1 is reported to promote mitochondrial function by stimulating cardiac mitochondrial function, increasing calcium influx in renal mitochondria [32], regulating mitochondrial dynamics [33], and maintaining mitochondrial bioenergetics and antioxidant response in endothelial [34] and renal tubular cells [35]. We now report that Stc1 is an miR 208a-dependent mitochondrial protective factor in cardiomyocytes exposed to metabolic challenges.

In our experiments, MED7 mRNA was decreased by diabetic conditions and normalized by the miR 208a suppression. MED7 is a component of the mediator complex, a multi-subunit link between RNA polymerase II and transcription factors to govern nuclear transcription [36]. Genetic alterations within the MED21–MED7 heterodimer impeded the binding of the mediator complex to RNA polymerase II [37]. MED7 mRNA has one conserved site in the 3′-UTR terminus, which can bind a nucleotide sequence in the miR 208a 3′ terminus located outside of the miR seed. MiR 208a was reported to repress the expression of another component of the mediator complex, MED13 (thyroid hormone-associated protein 1) [38], but we determined that, in human cardiomyocytes, MED13 mRNA is not an miR 208 target.

We conclude that miR 208a controls factors that maintain nuclear transcription, mitochondrial translation and bioenergetics, and mitochondrial and cellular antioxidant response.

### 4.3. Cardiac Stress Markers

Cardiac contractility is maintained by the balance between the fast ATPase α myosin isoform (MHC-α) encoded by the *myh6* gene and the slow β myosin isoform (MHC-β) encoded by the *myh7* gene. A hallmark of cardiac stress is the reactivation of a fetal genetic program, including upregulation of the atrial natriuretic peptide (ANP) and MHC-β, as well as downregulation of MHC-α (myosin switch) [39,40].

A primary decrease in *myh6* expression caused a decline in miR 208a, indicating that miR 208a follows the expression of its host gene during normal cardiac development [21]. We also found a positive correlation between miR 208a and the *myh6* product, MHC-α, in normal cardiomyocytes exposed to both HG and HP, but it is unclear if the decrease in *myh6* expression is the primary event leading to miR 208a deficiency during these diabetogenic conditions. Interestingly, the 90% miR 208a suppression corrected the metabolic-induced decrease in MHC-α. MiR 208a suppression also increased MHC-β in cardiomyocytes cultured in normal conditions. This inverse relationship between miR 208a and MHC-β in normal conditions contradicts the results observed in diabetic conditions where miR 208a deficiency blunted the MHC-β overexpression. Our results are in line with those reported in other models of cardiac disease. For example, in the pressure-overload stressed heart, miR 208a knockout blocked β-MHC protein expression [21].

In our model of cultured cardiomyocytes, both metabolic challenges caused a complete myosin switch, suggesting that metabolic stress may negatively affect contractility. Our results are in line with those observed in the diabetic human hearts [22] and show that miR 208a affects cardiac contractile proteins via a direct effect on cardiomyocytes rather than via its systemic extracardiac actions [23]. While a 30% decrease in miR 208a was not protective, a 90% miR 208a depression reversed the myosin switch and ANP overexpression, suggesting that miR 208a is not the sole regulator of cardiac contractility. A threefold increase in miR 208a is necessary to alter MHC-β, suggesting a threshold for the control of miR 208a on *myh-7* gene expression [21]. The negative effect of miR 208 on its targeted mRNA species is enhanced by stress [41] and potentially influenced by the type of cardiac stress. Because none of the cardiac stress marker mRNAs are direct miR 208a targets, miR 208a suppression may restore the myosin isoforms ratio via mechanisms other than directly regulating their mRNAs.

A candidate mechanism that may be responsible for this effect is the activation of thyroid hormone signaling. During cardiac development, thyroid hormone is an upstream positive regulator of both the host gene, *myh6*, and its intronic miR 208a [21], thus maintaining MHC-α and cardiac contractility [42]. In contrast, in the adult stressed heart, thyroid hormone signaling is reported downstream of miR 208a, and the effect of miR 208a on contractile proteins is dependent on thyroid hormone signaling [21]. MiR 208a deficiency enhanced thyroid hormone’s ability to inhibit *myh7* expression via one of its target genes, thyroid hormone-associated protein 1, also called MED13 [21]. MiR 208a repressed MED13 expression [38]. Therefore, miR 208a is a limiting factor for the thyroid hormone signaling on cardiac contractile proteins. We add to this concept the observation that miR 208a limits the thyroid hormone effect on contractile proteins via thyroid hormone receptor A.

In conclusion, consistent with other cardiac stress models [38], we confirm that miR 208a is a regulator of cardiac contractile proteins in short-term metabolically challenged cardiomyocytes. However, we cannot predict similar results in chronic metabolic conditions.

### 4.4. Mitochondrial Biogenesis

The mitochondrial biogenesis pathway, comprising, in a hierarchical manner, of PGC-1α, TFAM, and NRF1, is reportedly activated in hearts exposed to an HFD regimen [43]. We found that none of the mitochondrial biogenesis regulating factors were sensitive to miR 208a in normal conditions, an observation that is supported by our bioinformatics search that did not identify them as direct miR 208a targets. The 30% decrease in miR 208a induced by the metabolic stress was not protective against the decrease in some mitochondrial biogenesis factors, while 90% miR 208a suppression induced a robust mitochondrial biogenesis signal in cultured cardiomyocytes. In in vivo experiments with HFD-induced insulin resistant rodents, an increased mitochondrial FA oxidation matching the PGC-1α overexpression [44] aligned with the positive effect of FA on PGC-1α expression [45] and the concept that PGC-1α positively regulates FA β-oxidation enzymes and ETS subunits [46]. In contrast with the in vivo metabolic syndrome, metabolically challenged cardiomyocytes show a depressed mitochondrial biogenesis signal and a secondary decrease in mitochondrial ETS complexes. Strikingly, the total cardiomyocyte cytochrome c, a mitochondrial marker, was increased by the metabolic stress and miR 208 deficiency opposed this increase. MiR 208a restores the effect of HG and HP on downstream biogenesis factors, NRF and TFAM, and mitochondrial ETS complexes. Our results indicate that miR 208a opposes downstream positive regulators of mitochondrial biogenesis.

### 4.5. Mitochondrial Function

We observed that 30% miR 208a deficiency was insufficient to protect against the decrease in mitochondrial markers and mtDNA induced by the in vitro metabolic stress, whereas 90% miR 208a suppression was protective.

Although miR 208a status does not affect mitochondrial biogenesis, mtDNA and specific markers in normal conditions, both miR 208a deficiency and overexpression caused a slight decrease in basal oxygen consumption rates in cardiomyocytes energized with glucose and the complex I substrate pyruvate. This was associated with an increase in the extracellular acidification rates, suggesting that cultured cardiomyocytes trend to compensate for the decrease in mitochondrial function by increasing extramitochondrial glycolysis. Our data suggest that the miR 208a status specifically affects mitochondrial metabolism, while the extramitochondrial glycolytic ability is only secondarily affected. The maximal cardiomyocyte OCR is reached by achieving the ETS maximal capacity when the mitochondrial inner membrane proton gradient is collapsed by the uncoupler. While the basal OCR is slightly depressed, miR 208a suppression increases the reserve OCR, which is calculated by subtracting the basal from the maximal OCR. This feature is lost in cardiomyocytes overexpressing miR 208a, indicating that miR 208 is limiting for the maximal mitochondrial function.

Metabolically challenged cardiomyocytes maintain the basal and ADP-coupled OCR. In contrast, when the mitochondrial inner membrane proton gradient is collapsed with an uncoupler, miR 208a suppression enables ETS to work at its maximal capacity and increases maximal OCR. The data on mitochondrial function align with the increase in mitochondrial density and ETS subunits in miR 208a-deficient cardiomyocytes. The increase in mitochondrial maximal respiratory capacity provides a higher mitochondrial reserved function. These data are in agreement with the concept that the functional capacity of individual ETS subunits is higher than that needed for basal oxidative phosphorylation in cardiac mitochondria. Our data indicate that miR 208a limits this functional capacity by inhibiting mitochondrial biogenesis during metabolic stress.

### 4.6. Fatty-Acid β-Oxidation

MiR 208a participates in the angiotensin II signaling pathway to depress cardiac FA oxidation [47]. In our study, two major FA oxidation enzymes, CPT1 and LCAD, were not changed by the miR 208a suppression in cardiomyocytes cultured in normal conditions, confirming that they are not direct miR 208a targets. MiR 208a suppression canceled the effect of excessive substrates (both HG and HP) to decrease CPT1, indicating that miR 208a deficiency favors both CPT1-mediated FA transport and LCAD-mediated β-oxidation within the mitochondria. Our results are in contradiction with the effect of miR 208a on mitochondrial metabolism after a short-term (1 h) incubation of HL-1 cardiomyocytes with palmitate. Palmitate-supported respiration was higher than pyruvate-dependent respiration, and this difference was blunted by overexpressing miR 208a [47], suggesting that miR 208a may also have an acute control on mitochondrial metabolism via mechanisms other than gene expression. This statement is supported by the observation that, in that study, miR 208a expression did not affect nuclear receptors (PPARα protein) that regulate FA oxidation enzymes.

## 5. Conclusions

In conclusion, we report that miR 208 regulates cardiac mitochondrial biogenesis and function by acting on upstream and downstream regulators of mitochondrial translation, integrity, and protein turnover (Figure 6). Because the metabolic heart cannot freely switch between glucose and fatty-acid oxidation as energetic substrates, future research is needed to determine the effect of miR 208 on cardiac metabolic flexibility. Our data set the stage for future research to determine the benefit of downregulating miR 208a in protecting against organ disease in inherited and acquired mitochondrial dysfunctions.

## Figures and Tables

**Figure 1 cells-10-03152-f001:**
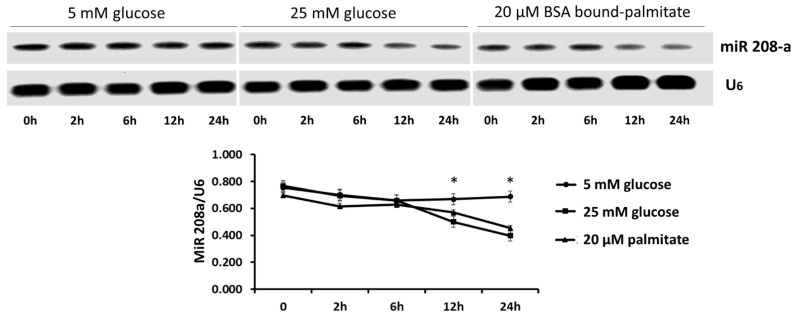
**MicroRNA 208a in human cardiomyocytes exposed to metabolic stress.** SV40 human cardiomyocytes were incubated with 5 mM glucose, 25 mM glucose or 20 mM bovine serum albumin (BSA)-bound palmitate for 0 to 24 h. The results are expressed as the mean ± SEM of three independent experiments. * *p* < 0.05 when comparing 5 mM glucose with either 25 mM glucose or 20 µM palmitate.

**Figure 2 cells-10-03152-f002:**
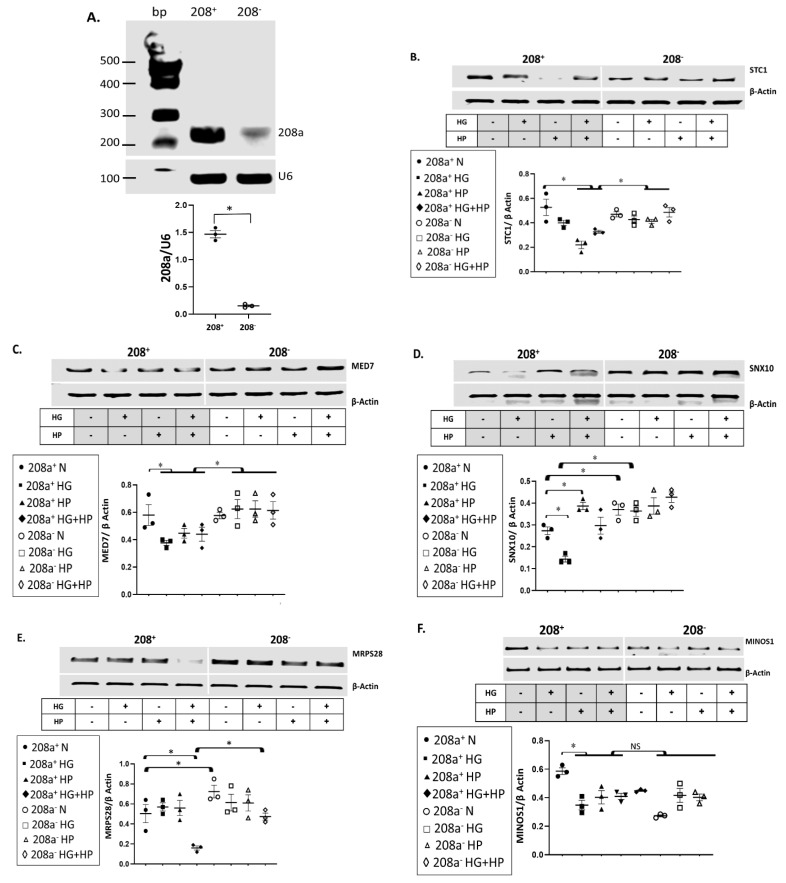
**mRNA targets of miR 208a in cardiomyocytes incubated in diabetogenic conditions.** (**A**) SV40 cardiomyocytes with normal (208^+^) and deficient (208^−^) miR 208a. First, 2 µg of total RNA from each sample was used to generate cDNA. RT-PCR cDNA products were used as templates for quantitative PCR (qPCR). All the qPCR assays were run in duplicate. RT-PCR products were separated on 1.5% agarose gels, while the bands were quantified and expressed relatively to the housekeeping mRNA *U6.* A Quick-Load^®^ 100 bp DNA Ladder was used as a molecular weight marker. (**B**) Stanniocalcin 1 (STC1). (**C**) Mediator complex subunit 7 (MED7). (**D**) Sortin nexin 10 (SNX10). (**E**) Mitochondrial ribosomal protein S28 (MRPS28). (**F**) Mitochondrial contact site and cristae organizing system protein 1 (MINOS1). SV40 human cardiomyocytes were incubated in DMEM with 5 mM glucose (N), 25 mM glucose (high glucose, HG), or 20 µM bovine serum albumin (BSA)-bound palmitate (high palmitate, HP) for 24 h. First, 2 µg of total RNA from each sample was used to generate cDNA. RT-PCR cDNA products were used as templates for quantitative PCR (qPCR). All the qPCR assays were run in duplicate. RT-PCR products were separated on 1.5% agarose gels, while the bands were quantified and expressed relatively to the housekeeping mRNA *β-actin.* The alignment between miR 208a sequence and specific mRNAs, as well as the level of conservation of those sequences in different species, is shown in Appendix A. Results are expressed as the mean ± SEM of three independent experiments. * *p* < 0.05.

**Figure 3 cells-10-03152-f003:**
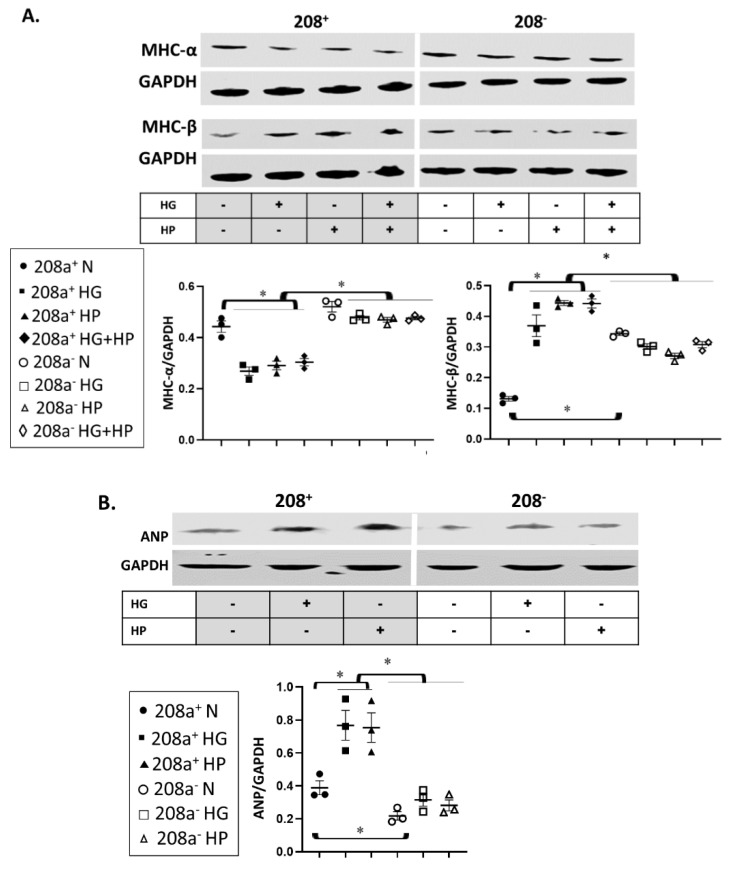
**MiR 208a regulates the cardiomyocyte stress response.** SV40 human cardiomyocytes were incubated in DMEM with 5 mM glucose, 25 mM glucose (high glucose, HG), or 20 µM bovine serum albumin (BSA)-bound palmitate (high palmitate, HP) for 24 h, and markers of pathologic cardiomyocyte hypertrophy were assessed. (**A**). MHC-α, myosin heavy chain-α; MHC-β, myosin heavy chain-β; GAPDH, glyceraldehyde 3-phosphate dehydrogenase. (**B**). ANP, atrial natriuretic peptide. The results are expressed as the mean ± SEM of three independent experiments. * *p* < 0.05.

**Figure 4 cells-10-03152-f004:**
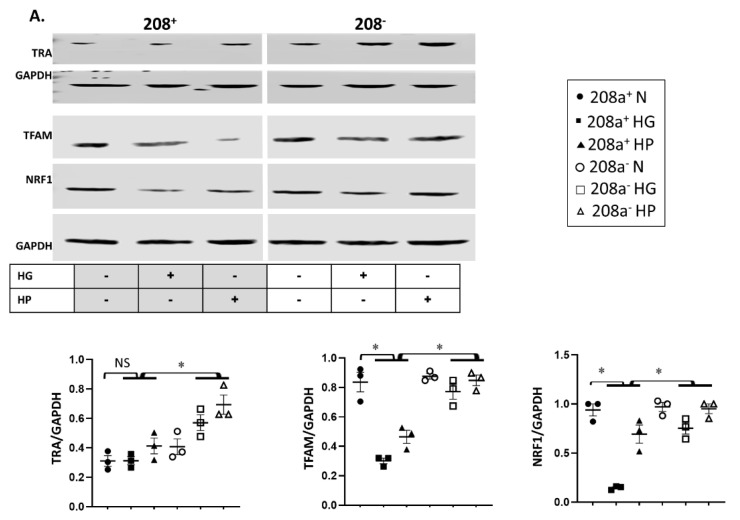
**MiR 208a, mitochondrial biogenesis, and mitochondrial markers.** SV40 human cardiomyocytes were incubated in DMEM with 5 mM glucose, 25 mM glucose (high glucose, HG), or 20 µM bovine serum albumin (BSA)-bound palmitate (high palmitate, HP) for 24 h. (**A**) Mitochondrial biogenesis factors. TRA, thyroid hormone receptor A; TFAM, mitochondrial transcription factor A; NRF1, nuclear respiratory factor 1. (**B**) PGC1 α, peroxisome proliferator-activated receptor γ-coactivator α. (**C**) Mitochondrial DNA. Total DNA was subjected to quantitative real-time PCR using a designed primer for mitochondrial NADH dehydrogenase 4 (ND4). (**D**) Mitochondrial marker proteins. C I, II, III, and V, complexes I, II, III, and V. (**E**) CPT1, carnitine palmitoyl transferase 1; LCAD, long-chain acylCoA dehydrogenase; GAPDH, glyceraldehyde 3-phosphate dehydrogenase. The lower panels represent the densitometric analyses of specific proteins per GAPDH. Results are expressed as the mean ± SEM of three independent experiments. * *p* < 0.05.

**Figure 5 cells-10-03152-f005:**
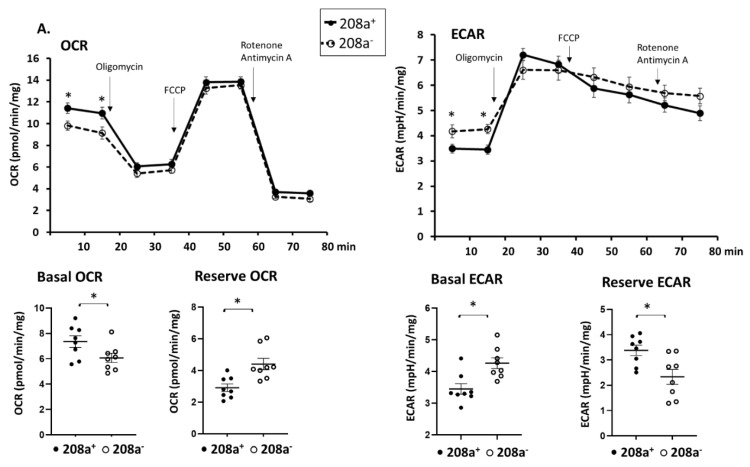
**MiR 208a deficiency protects against the decrease in mitochondrial function induced by diabetic conditions.** SV40 human cardiomyocytes were incubated in DMEM with 5 mM glucose, 25 mM glucose (high glucose, HG), or 20 µM bovine serum albumin (BSA)-bound palmitate (high palmitate, HP) for 24 h. Mitochondrial function was measured using an XFp Seahorse analyzer in bicarbonate- and phenol red-free DMEM supplemented with energetic substrates (10 mM glucose, 1 mM pyruvate, and 2 mM glutamine). (**A**) Respiratory properties induced by the miR 208a deficiency in cardiomyocytes exposed to normal conditions. (**B**) Respiratory properties induced by the miR 208a overexpression in cardiomyocytes exposed to normal conditions. (**C**) Respiratory properties of cardiomyocytes exposed to high-glucose (HG) conditions. (**D**) Respiratory properties of cardiomyocytes exposed to high-palmitate (HP) conditions. 208^+^, wildtype cardiomyocytes that normally express miR 208a; 208^−^, cardiomyocytes deficient in miR 208a; 208^++^, wildtype cardiomyocytes transfected with miR 208a; 208^−+^, cardiomyocytes deficient in miR 208a which were transfected with miR 208a. OCR, oxygen consumption rate (expressed as pmol/min/mg); ECAR, extracellular acidification rates (expressed as mpH/min/mg); FCCP (carbonyl cyanide-*p*-trifluoromethoxyphenylhydrazone). Respiratory parameters were measured or calculated as described in Section 2. The results are expressed as the mean ± SEM of four independent experiments. * *p* < 0.05.

**Figure 6 cells-10-03152-f006:**
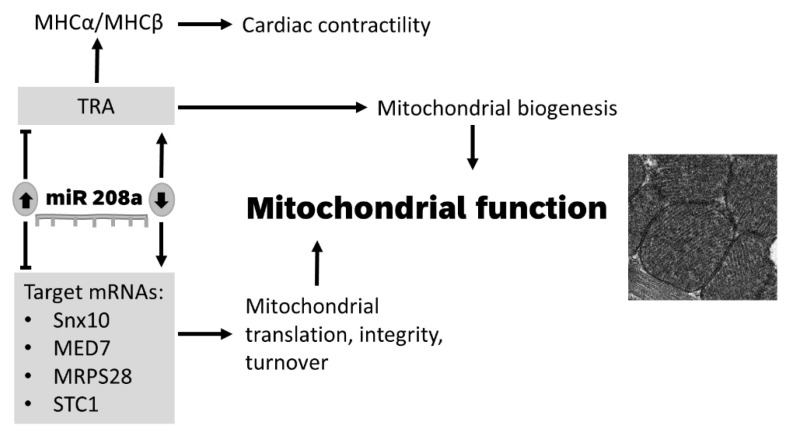
**Model for the role of miR 208a in regulating mitochondrial function in cardiomyocytes**. A decrease in miR 208a expression activates mitochondrial biogenesis via thyroid signaling and derepresses factors that control nuclear transcription, mitochondrial translation, oxidative stress, integrity, and turnover. SNX10, Sortin nexin 10; MED7, mediator complex subunit 7; MRPS28, mitochondrial ribosomal protein S28; STC1, Stanniocalcin 1; TRA, thyroid hormone receptor A; MHCα, myosin heavy chain α; MHCβ, myosin heavy chain β.

## Data Availability

Not applicable.

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
