# Peer review of "MiR 208a Regulates Mitochondrial Biogenesis in Metabolically Challenged Cardiomyocytes"

_cells, 2021, doi:10.3390/cells10113152_

Round 1
Reviewer 1 Report
The study of Mekala et al. addresses the impact of a metabolic challenges causing changes in regulation by MiR208a on mitochondrial transcription, function, and on control of mitochondrial biogenesis. Whereas it is an interesting topic and with very interesting data, the way it is presented makes it extremely hard to follow the idea.
Major comments:
- The abstract is difficult to read and understand without reading the paper.
- The downregulation of MiR 208a is achieved here by short term exposure of the cells to diabetic conditions (high glucose, high albumin-bound palmitate) and so it is impossible to clearly link the change in cardiac contractility and mitochondrial oxidative phosphorylation directly to the change in MiR 208a.
- The second paragraph of the introduction, lines 41-58, is extremely difficult to follow. It seems like the sentence are not following each other, and it is difficult to understand the idea that wants to be presented here.
- The paragraphs on lines 59-64 and the one on lines 65-72 are sort of lost the way they are presented. The flow in the text and the link between topics is not clear.
- Lines 110-111: The authors mentioned that the concentration of oligomycin and FCCP were optimized to obtain maximal effect. For the FCCP, this is normally done during the experiment, for each cells in the chamber, with a careful titration. This is not seen here, and it would be good to have more information on the optimization (was it try on each cell type, in each condition, etc.). Because it seems unlikely to me that the same concentration 5 μM was the optimal one in all conditions. The graphs seems to have two FCCP titration steps but it is not clear in the methods (is the 5 μM the results of the two titration steps, 2.5 μM each?).
- Some misalignments in the figures.
- In Fig. 5B, the OCR seems very similar with the graph of the trace, and there seems to be very large difference in the dot plot for the Basal OCR. I’m not sure I understand how so close data lead to such differences.
- In the results, the reading is not having a good flow and each section needs to have a summary of the general data meaning to see where this is going.
- The way the discussion is presented does not put their data at the first plan. E.g., in the paragraph lines 296-308, their data come only in the last 2 sentences, and the link with the previous sentences is not clear.
- The discussion section is extremely hard to follow and needs to be written more clearly, especially sections 4.1 and 4.2, and the last few paragraphs of 4.3.
- In the discussion, the presentation of the data on mitochondrial function is unclear in lines 432-438.
- As they brought at the end of the discussion the fatty acid beta-oxidation, it would have been interesting in their system to have the data of mitochondrial respiration in permeabilized cells, with substrates linked with glucose oxidation and substrates linked with fatty acid beta-oxidation.
Minor comments:
- Line 42-45: Sentence too long: ‘’ Pathogenic mechanisms leading to metabolic cardiomyopathy remain controversial, and include metabolic challenges (hyperglycemia and hyperlipidemia), insulin resistance and hyperinsulinemia, activation of the renin-angiotensin system, inflammation, epigenetic changes, oxidative stress, abnormalities in cardiac bioenergetics, all leading to cardiac re-modeling with hypertrophy, apoptosis and fibrosis [14]‘’.
- Line 46-48: Reference missing for the sentence.
- Line 49-50, sentence not clear: ‘’ As the major carbon fuels in both normal and metabolic heart, FA are 49 catabolized to acetyl-CoA feeding the tricarboxylic acid cycle (TCA) and electrons captured by the reducing equivalents, 50 NADH and FADH2, which are further oxidized by the electron transport chain (ETC) to form ATP‘’
- Electron transport chain (ETC) is not used anymore as it does not appear like a chain but like a system with several entry that converge to a linear pathway.
- ECAR is not clearly defined. The meaning of this measurement should be presented. It is an indicator of glycolysis by measuring the extracellular acidification rate.
- In figure 4, why is there no dot plot for panel C?
- 5, y axis, a closing parenthesis is missing after the unit and on some panels, units are completely missing.
- The sentence in line 315-316 is not clear: ‘’ We show 315 that a 30% miR 208a downregulation is insufficient to protect against bioenergetics deficit in cardiomyocytes incubated 316 in diabetogenic conditions whereas a 90% decrease is protective‘’.
- Not all abbreviations are define in the legend of Fig. 6.
Author Response
REVIEWER 1
The study of Mekala et al. addresses the impact of a metabolic challenges causing changes in regulation by MiR208a on mitochondrial transcription, function, and on control of mitochondrial biogenesis. Whereas it is an interesting topic and with very interesting data, the way it is presented makes it extremely hard to follow the idea.
We thank the reviewer for the appreciation of our work.
Major comments:
- The abstract is difficult to read and understand without reading the paper.
We modified the abstract so the readers can understand right away the major findings of our work.
- The downregulation of MiR 208a is achieved here by short term exposure of the cells to diabetic conditions (high glucose, high albumin-bound palmitate) and so it is impossible to clearly link the change in cardiac contractility and mitochondrial oxidative phosphorylation directly to the change in MiR 208a.
We agree with the reviewers. We have replaced the word “indicating” with “suggesting” in the abstract when referring to the causal link between miR 208a changes and the myosin heavy chain isoform expression. We also added a sentence in line 383 acknowledging that this is short-term study.
- The second paragraph of the introduction, lines 41-58, is extremely difficult to follow. It seems like the sentence are not following each other, and it is difficult to understand the idea that wants to be presented here.
We think that the paragraph needed a link sentence that must emphasize that from all pathogenic mechanisms, alterations in cardiac oxidative metabolism is critical. This was added on line 42.
- The paragraphs on lines 59-64 and the one on lines 65-72 are sort of lost the way they are presented. The flow in the text and the link between topics is not clear.
We have changed the entire paragraph to reflect the relationship between miR 208a and its host gene, myh6, in normal and pathological conditions.
- Lines 110-111: The authors mentioned that the concentration of oligomycin and FCCP were optimized to obtain maximal effect. For the FCCP, this is normally done during the experiment, for each cells in the chamber, with a careful titration. This is not seen here, and it would be good to have more information on the optimization (was it try on each cell type, in each condition, etc.). Because it seems unlikely to me that the same concentration 5 μM was the optimal one in all conditions. The graphs seems to have two FCCP titration steps but it is not clear in the methods (is the 5 μM the results of the two titration steps, 2.5 μM each?).
Seahorse XFp cartridges have only four injection ports and therefore allow only four injections in the well (size similar to the 96-well plate wells) during the experiment. Multiple titrations with FCCP are not possible during the experiment without compromising other additions such as oligomycin, Antimycin+Rotenone. This would be possible in the Oroboros chambers.
According to the protocol provided by the Seahorse Biosciences, we tried concentrations of Oligomycin 2.5-25 µM, and FCCP 2.5-10µM in normal and miR 208a-deficient cardiomyocytes cultured in normal conditions. We used the same range of concentrations in both types of cells because we did not observe differences in mitochondrial markers in normal conditions when comparing the 208+ versus 208- cells. We observed that for cardiomyocytes, 25 µM oligomycin maximally decreases the OCR. For FCCP, 2.5 µM increases the OCR, while 5 µM has no additional increasing effect, and 10 µM is inhibiting. However, different cells (ie, retinal photoreceptors, in our hands) require different optimal concentrations. Therefore, we chose oligomycin 25 µM and FCCP 5 µM as optimal concentrations.
We added more details in lines 112-113.
- Some misalignments in the figures.
We have tried to better align all the figures.
- In Fig. 5B, the OCR seems very similar with the graph of the trace, and there seems to be very large difference in the dot plot for the Basal OCR. I’m not sure I understand how so close data lead to such differences.
Basal OCR, as seen in the dot plot, is calculated by subtracting the non-mitochondrial OCR (minimal OCR after complete ETS inhibition with rotenone and antimycin A) from the OCR value prior to FCCP injection. This is why the individual values are lower and seem different from each other. As specified in the figure legend, the results are presented in Mean ± SEM in both (graph and dot plot), showing the individual data for each experiment in the dot plot figure.
- In the results, the reading is not having a good flow and each section needs to have a summary of the general data meaning to see where this is going.
We added a summary statement in all sections in the Results in order to emphasize the major findings.
- The way the discussion is presented does not put their data at the first plan. E.g., in the paragraph lines 296-308, their data come only in the last 2 sentences, and the link with the previous sentences is not clear.
We have modified all sections in the Discussion.
- The discussion section is extremely hard to follow and needs to be written more clearly, especially sections 4.1 and 4.2, and the last few paragraphs of 4.3.
We have changed completely these sections.
- In the discussion, the presentation of the data on mitochondrial function is unclear in lines 432-438.
We have changed these paragraphs.
- As they brought at the end of the discussion the fatty acid beta-oxidation, it would have been interesting in their system to have the data of mitochondrial respiration in permeabilized cells, with substrates linked with glucose oxidation and substrates linked with fatty acid beta-oxidation.
The scope of this study was to determine the effect of miR 208a om mitochondrial biogenesis and function. The reviewer is giving us a very good idea, this is a very interesting follow-up study, we thank the reviewer, and added it in the Conclusion section.
Minor comments:
- Line 42-45: Sentence too long: ‘’ Pathogenic mechanisms leading to metabolic cardiomyopathy remain controversial, and include metabolic challenges (hyperglycemia and hyperlipidemia), insulin resistance and hyperinsulinemia, activation of the renin-angiotensin system, inflammation, epigenetic changes, oxidative stress, abnormalities in cardiac bioenergetics, all leading to cardiac re-modeling with hypertrophy, apoptosis and fibrosis [14]‘’.
We have shortened the sentence (line 42).
- Line 46-48: Reference missing for the sentence.
This paragraph becomes now lines 45-50 in the new manuscript. We have considered that this is textbook normal biochemistry and did not add a reference.
- Line 49-50, sentence not clear: ‘’ As the major carbon fuels in both normal and metabolic heart, FA are 49 catabolized to acetyl-CoA feeding the tricarboxylic acid cycle (TCA) and electrons captured by the reducing equivalents, 50 NADH and FADH2, which are further oxidized by the electron transport chain (ETC) to form ATP‘’
We have clarified this sentence.
- Electron transport chain (ETC) is not used anymore as it does not appear like a chain but like a system with several entry that converge to a linear pathway.
We have changed the term to Electron Transport System (ETS) all over the manuscript.
- ECAR is not clearly defined. The meaning of this measurement should be presented. It is an indicator of glycolysis by measuring the extracellular acidification rate.
The definition of ECAR has been added in section 2.3. (line 121) as: Extramitochondrial glycolysis was assessed as extracellular acidification rate (ECAR).
- In figure 4, why is there no dot plot for panel C?
We corrected this deficiency.
- 5, y axis, a closing parenthesis is missing after the unit and on some panels, units are completely missing.
We corrected this deficiency.
- The sentence in line 315-316 is not clear: ‘’ We show 315 that a 30% miR 208a downregulation is insufficient to protect against bioenergetics deficit in cardiomyocytes incubated 316 in diabetogenic conditions whereas a 90% decrease is protective‘’.
We have changed completely that paragraph.
- Not all abbreviations are define in the legend of Fig. 6.
We have corrected this deficiency.

Reviewer 2 Report
Overall, this is an interesting paper that explores the impact of miR-208-a on mitochondrial physiology. The quality and care shown for the experiments is quite good, and the data are intriguing. Collectively, this is a nice body of work that starts to reveal the bioenergetic impact of miR-208-a.
The limitations of the work are primarily in the presentation of the figures. Overall, the resolution and organization can be improved to help with interpretation of the data. Specific comments are provided below:
- The overall resolution of Figure 2 is quite low. I would hope that this can be improved, especially for the alignments. The text is small and pixelated, making it har to assess. In fact, the alignments are not mentioned in the results section, so this could be moved to supplemental. This might make the figure less busy and the focus could be on the relative levels of mRNA levels.
- In Figure 2, the legend description of the different panels is off. You describe two B panels, so things got shifted.
- Figure 3 could be broken into three panels (i.e. A, B and C) for MHCs and ANP. The trends for alpha and beta are different, and the ANP is a different protein, so this would add clarity if this was better organized.
- The same issues arise in Figure 4. Several proteins are grouped together. And the positioning of the gene names chsanges from one panel to the next (left side or right?). This inconsistency is just distrating and makes it harder to interpret the results.
- In Figure 5, the X-axis of the quantified respiration measurements is missing. I know that the symbols are meant to indicate, but the condition can easily be indicated on the graph. And this would be consistent with the other parts of the figure. Additionally, TG is never explained in the text or the figure legend. I think this is adding back the 208a, but different from the Western.
- The Discussion at times focused on restating results with little interpretation. This section could be shorter and focus on the impact of their findings.
- The increase in cytochrome c is interesting. Is this because of the increase in apoptosis? Is the cytochrome c level reflect the extra-mitochondrial protein that has leaked into the cytosol. This would be interesting to check by isolating mitochondrial fractions to see if mitochondrial cyt c levels change. Moreover, the potential meaning of this observation is not discussed in the Discussion.
Minor points...
- More specific details of reagents used could/should be provided.
- Line 162 - "...a time-dependent up to 30% decrease in miR 208-a expression." Awkward wording. Maybe "...a time-dependent decrease, up to 30%, in miR-208-a expression."
Author Response
REVIEWER 2
Overall, this is an interesting paper that explores the impact of miR-208-a on mitochondrial physiology. The quality and care shown for the experiments is quite good, and the data are intriguing. Collectively, this is a nice body of work that starts to reveal the bioenergetic impact of miR-208-a.
We thank the reviewer for appreciating our work.
The limitations of the work are primarily in the presentation of the figures. Overall, the resolution and organization can be improved to help with interpretation of the data.
We have replaced all figures with figures with better resolution.
Specific comments are provided below:
- The overall resolution of Figure 2 is quite low. I would hope that this can be improved, especially for the alignments. The text is small and pixelated, making it har to assess. In fact, the alignments are not mentioned in the results section, so this could be moved to supplemental. This might make the figure less busy and the focus could be on the relative levels of mRNA levels.
We agree with the reviewer, and moved these parts of Figure 2 in the Supplemental figures.
- In Figure 2, the legend description of the different panels is off. You describe two B panels, so things got shifted.
We apologize for this error, and corrected it.
- Figure 3 could be broken into three panels (i.e. A, B and C) for MHCs and ANP. The trends for alpha and beta are different, and the ANP is a different protein, so this would add clarity if this was better organized.
We divided Figure 3 as suggested by the reviewer.
- The same issues arise in Figure 4. Several proteins are grouped together. And the positioning of the gene names chsanges from one panel to the next (left side or right?). This inconsistency is just distrating and makes it harder to interpret the results.
We also divided Figure 4 as suggested by the reviewer.
- In Figure 5, the X-axis of the quantified respiration measurements is missing. I know that the symbols are meant to indicate, but the condition can easily be indicated on the graph. And this would be consistent with the other parts of the figure. Additionally, TG is never explained in the text or the figure legend. I think this is adding back the 208a, but different from the Western.
We think that by adding text on the x-axis will make the Figures very busy. However, at the suggestion of the reviewer, we have tried to make the figure easier to understand.
- The Discussion at times focused on restating results with little interpretation. This section could be shorter and focus on the impact of their findings.
We have changed the whole Discussion sections.
- The increase in cytochrome c is interesting. Is this because of the increase in apoptosis? Is the cytochrome c level reflect the extra-mitochondrial protein that has leaked into the cytosol. This would be interesting to check by isolating mitochondrial fractions to see if mitochondrial cyt c levels change. Moreover, the potential meaning of this observation is not discussed in the Discussion.
We added in the text that the level of cytochrome c reflects the whole cellular cytochrome c, and is assessed as a mitochondrial marker. The goal of our study was to determine the effect of miR 208a on mitochondrial function rather than its involvement in apoptosis. However, this is a very interesting idea that may be investigated in future research.
Minor points...
- More specific details of reagents used could/should be provided.
We have tried to add details of the reagents within the method section.
- Line 162 - "...a time-dependent up to 30% decrease in miR 208-a expression." Awkward wording. Maybe "...a time-dependent decrease, up to 30%, in miR-208-a expression."
We fixed this odd way to express the results.

Round 2
Reviewer 1 Report
The changes made by the author are sufficient to adress my concerns.